# Technical and Environmental Feasibilities of the Commercial Production of NaOH from Brine by Means of an Integrated EDBM and Evaporation Process

**DOI:** 10.3390/membranes12090885

**Published:** 2022-09-14

**Authors:** Marta Herrero-Gonzalez, Raquel Ibañez

**Affiliations:** Departamento de Ingenierías Química y Biomolecular, Universidad de Cantabria, Avenida de los Castros s/n, 39005 Santander, Cantabria, Spain

**Keywords:** NaOH, brine, electrodialysis, bipolar membranes, evaporation, circular economy

## Abstract

Electrodialysis with bipolar membranes (EDBMs) is a technology that offers a great potential for the introduction of the principles of a circular economy in the desalination industry, by providing a strategy for the recovery of HCl and NaOH from brine via the process of seawater reverse osmosis (SWRO). Both chemicals are widely employed in desalination facilities, however NaOH presents a special interest due to its higher requirements and cost. Nevertheless, the standard commercial concentrations that are commonly employed in the facilities cannot be obtained using the state of the art EDBM technology itself. Therefore, the aim and main purpose of this work is to prove the technical and environmental feasibilities of a new approach to produce commercial NaOH (50%wt.) from SWRO brine by means of an integrated process of EDBMs followed by a triple effect evaporation. The global process has been technically evaluated in terms of the specific energy consumption (SEC) (kWh·kg^−1^ NaOH) and the environmental sustainability performance has been analyzed by its carbon footprint (CF) (kg CO_2_-eq.·kg^−1^ NaOH). The influence of the current density, and the power source in the EDBM stage have been evaluated on a laboratory scale while the influence of the feed stream concentration in the evaporation stage has been obtained through simulations using Aspen Plus. The lowest SEC of the integrated process (SEC_OV_), 31.1 kWh·kg^−1^ NaOH, is obtained when an average current density of 500 A·m^−2^, provided by a power supply (grid mix), is applied in the EDBM stage. The environmental burdens of the integrated process have been quantified by achieving reductions in the CF by up to 54.7% when solar photovoltaic energy is employed as the power source for EDBMs, with a value of 5.38 kg CO_2_-eq.·kg^−1^ NaOH. This study presents a great potential for the introduction of the principles of a circular economy in the water industry through the recovery of NaOH from the high salinity waste stream generated in SWRO facilities and opens the possibility of the reuse of NaOH by its self-supply in the desalination plant.

## 1. Introduction

The worldwide desalination capacity reached 95 million m^3^·day^−1^ in 2019, which saw a huge rise caused by the development of the reverse osmosis (RO) technology that currently makes up 65.5% of the desalination market, and in particular by seawater reverse osmosis (SWRO) (34% of the total desalination capacity) [1]. Despite its superior development, SWRO has two major environmental drawbacks [2,3]: (i) high-specific energy requirement (2.5–4.0 kWh·m^−3^ of freshwater [4]), which contributes to indirect greenhouse gas emissions due to the use of conventional energies based on fossil fuels, and (ii) direct brine (hypersaline waste stream) disposal into bodies of water, which harms the marine ecosystems [5,6,7,8].

The proposals made to reduce the high energy consumption focus on energy-saving technologies and equipment, being able to achieve energy consumptions close to the thermodynamic limit and later on, the integration of renewable energies, such as solar photovoltaic (PV) or wind energy, in order to provide a better environmental performance [4,9,10].

The initial strategies put forward to minimize the impact of brine discharge were the development of more efficient discharge systems to facilitate the dilution of the brine itself in the receiving media [2,11], and the minimization of the volume of brine to be disposed, and most of these focused on the evaporation of water [12]. The most recent proposals are focused on the material and/or energy recovery; thus, brine is no longer considered a waste stream but a source of valuable resources. In this way, brine can be reintroduced into the production process, which, from a circular economy point of view, presents potential environmental and economic advantages. Additionally, the adaptation of these processes, from a circular economy perspective, is encouraged by various European programs [13,14]. Several materials can be recovered or produced from brine, among them: (i) different salts, such as magnesium hydroxide, sodium and calcium chloride, calcium carbonate, or sodium sulphate [15], (ii) metals [16,17,18], many of these, with their conventional supply, are considered to be critical, and (iii) acids and bases, such as HCl and NaOH, that can be produced on-site using the electrodialysis with bipolar membrane (EDBM) technology, so that the requirements of these products in the desalination facility could be completely or, at least partially covered.

Furthermore, energy can be harvested from brine, through the salinity gradient (difference in the salt concentrations between two streams) [19] and the pH gradient (difference in the proton concentrations, pHs, between two streams) [20] either alone or combined [21].

In this sense, different technologies can be employed for brine management and valorization by itself. However, remarkable research efforts have been carried out in the last few years regarding the development of integrated systems which combine different technologies, the so called minimum liquid discharge (MLD) and zero liquid discharge (ZLD) systems, with the aim of avoiding or at least minimizing the volume of liquid waste streams that are generated and discharged into the environment [22,23,24,25].

Both HCl and NaOH are highly consumed in SWRO facilities, e.g., a SWRO plant located in the Mediterranean area requires doses of HCl of 0.2–0.5 g·m^−3^ of freshwater and of NaOH of 30–60 g·m^−3^ of freshwater, both employed at commercial standard concentrations, 31–35%wt. and 50%wt., respectively [26]. Concentrations below commercial standards could be employed, however, it could incur greater pumping requirements and, therefore, costs. These high reagent requirements, even as commodities, can lead to greater expenses for the SWRO facilities due to their prices, 93.75 EUR·ton^−1^ for the HCl 35%wt. and 196.43 EUR·ton^−1^ for the NaOH 50%wt. [27].

The literature shows the increasing relevance of the recovery of NaOH and HCl by different technologies such as electrodialysis, EDBMs, or electrosynthesis [28,29].

The EDBM technology is able to produce HCl and NaOH from brine and electric energy. The literature commonly reports concentrations lower than 2.0 mol·L^−1^ for both HCl and NaOH [30,31,32,33,34,35,36], which limits the integration of this technology within the desalination industry due to the large volume of freshwater (main desalination product) that is required in the production. Nevertheless, concentrations of up to 3.3 mol·L^−1^ (~10.2%) of HCl and 3.6 mol·L^−1^ (~9.7%) of NaOH can be obtained by means of EDBMs from brine using this technology in its current state [26]. Similar reagent concentrations can be achieved if a renewable energy source such as PV solar energy is employed as the power supply [26,37] instead of the grid mix, which provides a better environmental performance [38,39]. These concentrations are still far from acceptable commercial standards, so further concentration stages should be considered in order to avoid any technical disadvantages due to the increase in the pumping requirements or the potential storage of large volumes of products.

Previous studies by the research group were focused on the analysis through the simulation of the HCl concentration stage, using as feed, a diluted form of HCl, experimentally obtained in an EDBM process from SWRO brine and powered by a grid mix (constant current density) and PV solar energy (variable current density) [26]. An azeotropic distillation of the mixture HCl/H_2_O was simulated using the Aspen Plus [40] software. The influence of the HCl feed stream concentration (0.5 mol·L^−1^ to 3.0 mol·L^−1^), i.e., the output of the EDBM, in the specific energy consumption (SEC), from both the distillation and the integrated EDBM + distillation processes were evaluated. Different scenarios were considered for the EDBM process (experimentally obtained data), which differed in the average current density and the power supply source selected, and both variables conditioned the EDBM output product concentrations.

Despite the fact that the production of the commercial standard concentration of HCl has been evaluated, there is a need to provide strategies to obtain commercial or at least highly concentrated solutions of NaOH from highly salty waste streams. Moreover, the NaOH recovery presents a great interest due to its higher consumption in SWRO facilities and higher market prices, compared with HCl.

Thus, the aim of the present study is to contribute to the development of new strategies that integrate the membrane processes with the complementary stages based in sustainability principles, by presenting for the first time in the literature, the performance of an integrated process including EDBMs and evaporation stages fed by SWRO concentrates, in order to achieve the commercial concentrations of NaOH (50%wt). The evaluation is carried out through an analysis of the energetic performance by means of the SEC of the evaporation (SEC_EV_) and the integrated EDBMs and evaporation process (SEC_OV_), and the environmental performance by means of the carbon footprint (CF).

## 2. Materials and Methods

In this work, a novel integrated process of EDBMs combined with evaporation is proposed in order to produce NaOH at standard commercial concentrations, is evaluated. The SWRO brine is fed to the EDBM unit where NaOH and HCl are produced, then the NaOH is concentrated up to the commercial concentration in a triple effect evaporation. This study was carried out using laboratory experimental work for the EDBMs and through simulations using Aspen Plus for the evaporation. Therefore, in this section a description of the methodology for the EDBM experiments and the simulation of the triple effect evaporation is presented, together with the procedure of the SEC_EV_, SEC_OV,_ and CF_OV_ calculations.

### 2.1. EDBM Experimental Results

The experiments were performed in a modified PCCell (Germany) bench scale laboratory ED system, using a cell composed of two electrodes made of titanium and coated with ruthenium oxide and an effective area of 100 cm^2^, assembled with a triplet of membranes (anion, cation, and bipolar) and two cation membranes to create the electrolyte compartment. The commercial membranes AM-PP RALEX as anion and CM-PP RALEX as cation from Mega (Czech Republic), and Fumasep FBM from Fumatech (Germany) were selected.

A 20 L quantity of a simplified synthetic model SWRO brine (1.0 mol·L^−1^ NaCl), 1 L of 0.1 mol·L^−1^ HCl, and 1 L of 0.1 mol·L^−1^ NaOH were employed as initial solutions.

The performance of the EDBM technology under the constant current density, using a power supply (Statron, Germany) supported by the grid mix, and the variable current density, using a PV solar array simulator (Chroma, Ede, The Netherlands) that emulated the use of the PV panels, was analyzed considering three average current densities 500 A·m^−2^, 750 A·m^−2^, and 1000 A·m^−2^.

When the constant current densities are employed, these will take the value of the average current density 500 A·m^−2^, 750 A·m^−2^, and 1000 A·m^−2^. However, the case of the variable current density is more complex.

The PV solar array simulator emulates the performance of a PV panel by introducing an irradiation profile and the panel parameters. In this study, the same conditions as those used for the previous studies are employed [26,37]. An average irradiation profile for the month of July in Almeria (summer in South East Spain) was obtained from the PV-GIS database [41]. The obtention of a specific average current density was accomplished by sizing the simulated panel area, and this area was calculated in proportion with the previous experimental results [26,37].

A more detailed description of both the experimental set-up and procedure was explained in the previous studies [26].

### 2.2. Simulation of a Triple Effect Evaporation

In order to obtain NaOH at commercial standard concentrations (50%wt.), the simulation of the NaOH concentration stage, by means of evaporation after a first stage of the EDBM, was carried out.

Aspen Plus [40] was used as a software tool. A triple effect evaporation and ELECNRTL method were selected to simulate the electrolytes involved in this process. ELECNRTL calculates the properties of the liquid phase through the Electrolyte-NRTL activity coefficient model, while those of the vapor phase are calculated with the Redlich–Kwong equation of state. This method allows for the modeling by using real components (ionic species) that are generated with the initially specified components. The reactions (equilibria, precipitation, and dissociation) between these components in the liquid phase are also generated.

As depicted on Figure 1, a countercurrent flow configuration, i.e., a steam is fed in the first effect and the diluted NaOH is fed in the third effect, has been selected. This is in agreement with the literature [42,43,44].

The heat exchange in an evaporator can be produced isothermally (constant temperature) or isobarically (constant pressure), in this simulation, the constant pressure has been selected, with pressures of 1.00 atm, 0.75 atm, and 0.50 atm for the first, second, and third effects, respectively.

The feed stream is introduced in the third effect, using the liquid output of this effect as feed for the previous effect and so on, successively until the final product (NaOH 50%wt.) is obtained as the liquid output of the first effect.

Moreover, the first effect heat transfer is carried out by a saturated industrial steam stream at 200 °C. The second and third effects are fed with the vapor streams produced in the previous first and second effects, respectively. The diluted NaOH from the EDBMs is considered to be fed at 20 °C, a temperature that could fit in the EDBM outlet without heating or refrigeration between the EDBMs and the evaporation processes.

In this way, the system has two inputs, feed (NaOH from EDBM) and steam, while it has five outputs, the targeted product (NaOH 50%wt.), the steam from the third effect, and the condensates (considered as 100% water) from the three effects. Thus, in addition to the targeted product, water that can be reused and steam that can be recompressed for later use, are obtained.

Focusing on the simulation, the required input data is the concentration of the NaOH from EDBM stream, considering a range of 7–10%wt. (2.6–3.8 mol·L^−1^), which includes the concentrations of NaOH obtained by means of the EDBMs in previous experimental studies [26]. The production of NaOH at 50%wt. and the requirements of the industrial steam are calculated as results.

### 2.3. SEC Calculations

The SEC of the EDBM unit (SEC_EDBM_) is calculated on the basis of the NaOH (target product) produced, through Equation (1):(1)$${\mathrm{SEC}}_{\mathrm{EDBM}}\left(\frac{\mathrm{kWh}}{\mathrm{kg}\;\mathrm{NaOH}\;\mathrm{pure}}\right)=\frac{\int_{t=0}^{{t=t}_{e}}U\cdot I\cdot \mathrm{dt}}{{(C}_{{\mathrm{NaOH},t}_{e}}{-C}_{\mathrm{NaOH},0}{)\cdot V}_{\mathrm{NaOH}}{\cdot \mathrm{PM}}_{\mathrm{NaOH}}}$$
where S and F are the mass flows of the steam and the feed, respectively, H_S_ is the enthalpy of the steam stream, and x_F_ is the mass fraction of NaOH in the feed stream. where U is the total stack voltage (V), I is the current intensity (A), C_NaOH,te_ and C_NaOH,0_ are the concentrations of NaOH at the end and at the beginning of the experiment (mol·L^−1^), V_NaOH_ is the volume of NaOH (L), PM_NaOH_ is the molecular weight of NaOH (g·mol^−1^), t_e_ is the total time of the experiment (h), and t is time (h).

Although NaOH is produced together with the HCl, in this study, the SEC_EDBM_ is referred to as a unit of mass of NaOH as it is considered the target product whereas the HCl is a byproduct.

Moreover, the SEC of the evaporation (SEC_EV_) is calculated from the concentration of the NaOH from the EDBM stream (input to evaporation) and from the requirements of the industrial steam by applying the expression in Equation (2):(2)$${\mathrm{SEC}}_{\mathrm{EV}}\left(\frac{\mathrm{kWh}}{\mathrm{kg}\;\mathrm{NaOH}\;\mathrm{pure}}\right)=\frac{S\left(\frac{\mathrm{kg}\;\mathrm{steam}}{h}\right){\cdot H}_{S}\left(\frac{\mathrm{kJ}}{\mathrm{kg}\;\mathrm{steam}}\right)}{F\left(\frac{\mathrm{kg}\;\mathrm{feed}}{h}\right){\cdot x}_{F}}\cdot \frac{1\;(\mathrm{kWh})}{3600\;(\mathrm{kJ})}$$
where S and F are the mass flows of the steam and the feed, respectively, H_S_ is the enthalpy of the steam stream, and x_F_ is the mass fraction of NaOH in the feed stream.

The evaluation of the integrated process EDBM + Evaporation would be carried out through the calculation of the SEC of the overall process (SEC_OV_), that can be calculated as the sum of the SEC of the EDBM process (SEC_EDBM_) and the SEC_EV_ (Equation (3)):(3)$${\mathrm{SEC}}_{\mathrm{OV}\;}{=\mathrm{SEC}}_{\mathrm{EDBM}}+{\mathrm{SEC}}_{\mathrm{EV}}$$

### 2.4. CF Calculations

The environmental performance of the integrated process of the EDBMs coupled with the evaporation could be assessed through the calculation of the carbon footprint (CF) of the overall energy input to the system. In this sense, the CF is calculated according to Equation (4):(4)$${\mathrm{CF}}_{\mathrm{OV}}{=\mathrm{CF}}_{\mathrm{EDBM}}+{\mathrm{CF}}_{\mathrm{EV}}={\mathrm{SEC}}_{\mathrm{EDBM}\;}\cdot {\;\mathrm{CF}}_{E}+{\mathrm{SEC}}_{\mathrm{EV}\;}\cdot {\mathrm{CF}}_{S}$$
where CF_E_ is the CF of the energy input, if the grid mix (Spanish, composed by a 36% of renewable energies [45]) is considered, thus a value of 0.338 kg CO_2_-eq.·kWh^−1^ will be obtained, whereas a value of 0.036 kg CO_2_-eq.·kWh^−1^ will be achieved if the solar PV energy is selected [39], and CF_S_ is the CF for the steam (0.294 kg CO_2_-eq.·kWh^−1^) [46].

## 3. Results and Discussion

The results from the simulation of the triple effect evaporation carried out using the Aspen Plus software, together with the SEC_EV_, SEC_OV,_ and CF_OV_ calculations, are presented.

### 3.1. EDBM Experimental Results

The experimental study reports the production of HCl and NaOH by means of EDBMs from brine (1.0 mol·L^−1^ NaCl), with concentrations up to 3.3 mol·L^−1^ (~10.2%) of HCl and 3.6 mol·L^−1^ (~9.7%) [26]. Table 1 presents a summary of the experimental results focusing on the NaOH production under three different average current densities 500, 750, and 1000 A·m^−2^, and under different power sources, a grid mix (constant current density) and solar PV energy (variable current density).

As seen in Table 1, the increase in the average current density, in both the grid mix and solar PV energy, increases the maximum NaOH concentration achieved, however, the SEC_EDBM_ is also increased. Moreover, similar NaOH concentrations and slightly a higher SEC_EDBM_ are obtained when solar PV energy is selected as the power source instead of the grid mix.

Thus, the integration of EDBMs with evaporation with the target of producing NaOH at commercial concentrations (50%wt.) is evaluated in terms of the SEC. In this sense, the feed NaOH concentrations to the evaporation in the range of 7–10%wt. are studied, which correspond with the maximum concentration experimentally achieved under different conditions.

### 3.2. Simulation Results of a Triple Effect NaOH Evaporation Process

Figure 2 depicts the requirements of the industrial steam and the production of NaOH at the targeted concentration (50%wt.), obtained from the simulation, for a feed with concentrations on the range of 7–10%wt. (2.6–3.8 mol·L^−1^). As expected, an increase in the feed concentration favors the evaporation process in two ways: (i) industrial steam requirements are reduced, and (ii) the production of NaOH 50%wt. is increased. The production of NaOH 50%wt. is increased in the same proportion as the feed concentration is increased, whereas the industrial steam requirements are not proportional.

Increasing by ~43% the NaOH feed concentration (from 7 to 10%) implies a reduction of 3.5% of the requirements of the industrial steam. Likewise, the water recovery will be reduced if the feed concentration is increased, as less water (in terms of proportion) would be fed to the evaporation system.

### 3.3. SEC Results

The SEC_EDBM_ has been updated from previous experimental results [26], considering a NaOH dry base (pure) instead of HCl. Although in the EDBMs, HCl and NaOH are produced simultaneously, a split of the SEC_EDBM_ between the two products has not been considered, as they are attributed completely to the NaOH as it is the target product due to its higher requirements in the SWRO facility and cost.

Table 2 reports the SEC_EDBM_ values for different NaOH concentrations achieved in the EDBM process (EDBM output), thus, a comparison of the profiles of the different current densities can be carried out without depending on the maximum concentration achieved for the product. In addition, the SEC_EV_ values for the different NaOH input concentrations to the evaporation system are also included in Table 2 and Figure 3. As expected and previously reported for the HCl [26], as the higher product concentrations are achieved in the EDBM unit, higher SEC_EDBM_ values are required. Moreover, the SEC_EDBM_ increases if higher average current densities are employed, with increments in the range of 0.79–2.31 kWh·kg^−1^ of NaOH (dry base) for every 1.0%wt. of the NaOH increase in the concentration at the output stream of the EDBMs.

Additionally, as higher concentrations than those commonly found in the literature are reported, the influence of non-ideal phenomena such as concentration polarization, proton leakage, or counter diffusion points out, which leads to a higher SEC_EDBM_ for the same product concentration (e.g., 7%wt.) when operated with higher average current densities, even if the operation times can be reduce. However, increasing the average current density is required in order to obtain more concentrated products.

As seen in equation 2, the SEC_EV_ is not only influenced by the steam requirements, but also by the production. Therefore, both factors will simultaneously favor its reduction. As disclosed in Table 2 and Figure 3, increasing the feed concentration from 7% to 10% NaOH (~43%) reduces the SEC_EV_ in a 32.4%, nevertheless, this reduction is not linear, as it shows 13.5% for the range of 7–8%, 10.5% for 8–9%, and 8.4% for 9–10%, which means that as the input is concentrated, the impact of the reduction of the SEC_EV_ is smaller because the requirements for steam remain high.

Figure 4 shows the SEC_OV_ for the integrated process of EDBMs and evaporation in order to obtain NaOH at commercial concentrations (50%wt.), unachievable with EDBMs alone. The higher average current densities report a higher SEC_OV_, nevertheless, given the same SEC_OV_, operating at higher average current densities allows for reaching higher concentrations. moreover, increasing the input concentration, within the same operating conditions, allows the reduction of the SEC_OV_ in a range of 6.2–13.8%, due to a rise in the contribution of the EDBM in the SEC_OV_ from a 34–47% at 7%wt. NaOH to a 45–59% at 10%wt. NaOH. Under the experimental conditions used, the lowest SEC_OV_ is achieved in the scenario G500, Figure 4, i.e., with an average current density of 500 A·m^−2^ and using the grid mix (constant current density) as an electrical power source, especially if the EDBM achieves the maximum NaOH concentrations under these conditions (~7.8%wt. NaOH).

### 3.4. CF Results

Even though, by using solar PV energy as an electrical power source requires a SEC_OV_ similar or slightly higher than using grid mix, it can present a better environmental performance. Renewable energies, in particular solar PV energy, present a lower CF than the grid mix, that is commonly based on fossil fuels. As an example, the CF of the current Spanish grid mix can be estimated to 0.338 kg CO_2_-eq.·kWh^−1^, whereas the CF of solar PV energy is 0.036 kg CO_2_-eq.·kWh^−1^, which is an order of magnitude lower [39]. Otherwise, the CF of industrial steam is considered 0.294 kg CO_2_-eq.·kWh^−1^ [46].

As plotted on Figure 5, the CF_OV_ provides values between 5.38 kg CO_2_-eq.·kg^−1^ NaOH (9.5%wt. and PV1000 + Evaporation) and 13.27 kg CO_2_-eq.·kg^−1^ NaOH (7.0%wt. and G1000 + Evaporation). As with the SEC_OV_, increasing the evaporation inlet concentration decreases the CF_OV_ by up to 24.5%. Increasing the average current density increases the CF_OV_ when operating with the grid mix (constant current density), while operating with solar PV energy, the CF_OV_ are very similar, only slightly lower at the higher average current densities. Moreover, replacing the grid mix with solar PV energy allows reductions of up to 38.3%, 41.0%, and 54.7% for the average current densities of 500, 750, and 1000 A.·m^−2^, respectively.

Therefore, from an environmental point of view, the best performance is obtained from the scenario PV1000 + Evaporation (9.5%wt.) with a CF_OV_ of 5.38 kg CO_2_-eq.·kg^−1^. However, benchmarking against the production of NaOH as reported in the Ecoinvent [46] database, based on the conventional production of NaOH (1.32 kg CO_2_-eq.·kg^−1^ NaOH) does give lower values than using solar PV energy and steam. If it is assumed that the steam used in the distillation stage is produced by the PV solar energy, the CF_OV_ would be reduced to 1.33 kg CO_2_-eq.·kg^−1^ NaOH.

However, it must be considered that the comparison does not consider the allocation procedure in the EDBM unit in which HCl is also generated as a coproduct, while the value given by the Ecoinvent database does not include transportation from the NaOH production site to the desalination facility. Taking both factors into account could bring the values of our proposal closer to the conventional one.

## 4. Conclusions

The viability of the production of NaOH at standard commercial concentrations (50%wt.) from SWRO brine by an integrated process that combines the EDBM technology and evaporation has been demonstrated and evaluated in terms of the SEC and CF. The proposed integrated process encourages the valorization of salty waste streams from the water industry and offers a potential strategy for the application of circular economy principles in the desalination industry.

The technical and environmental feasibilities has been evaluated from the experimental (EDBM) and the simulation (evaporation) results. The SEC_EDBM_ values in the range of 11.2 and 21.5 kWh·kg^−1^ of NaOH are obtained under the different conditions studied. Despite the fact that selecting higher average current densities lead to higher concentrations and a reduction of freshwater consumption, it also leads to a higher SEC_EDBM_. The SEC_EV_ can be reduced by up to 32.4% when the NaOH solutions with higher concentrations (~10%wt.) are fed to the evaporation.

The global process generates SEC_OV_ values between 31.1 and 42.1 kWh·kg^−1^ of NaOH. The SEC_OV_ is reduced when low average current densities are employed in the EDBM stage, although the achieved NaOH concentrations are lower (~7%wt.).

Selecting solar PV energy instead of the grid mix as a power source for the EDBMs slightly increases the SEC_OV_, however, it provides a better environmental performance. The minimum value of CF_OV_ of 5.38 kg CO_2_-eq.·kg^−1^ NaOH is achieved under solar PV energy and 1000 A·m^−2^, which means a reduction of up to 54.7%. Even though this value is improved when a renewable power source is employed in the EDBMs, it is still far from the value reported by the Ecoinvent database for the conventional production process of NaOH (1.32 kg CO_2_-eq.·kg^−1^ NaOH), which does not consider the transportation of the product from the production site to the desalination facility.

This work contributes to the introduction of circular economy principles in the water industry through the recovery of NaOH from the high salinity waste stream generated in SWRO facilities and opens up the possibility of the reuse of NaOH through self-supply in the desalination plant.

## Figures and Tables

**Figure 1 membranes-12-00885-f001:**
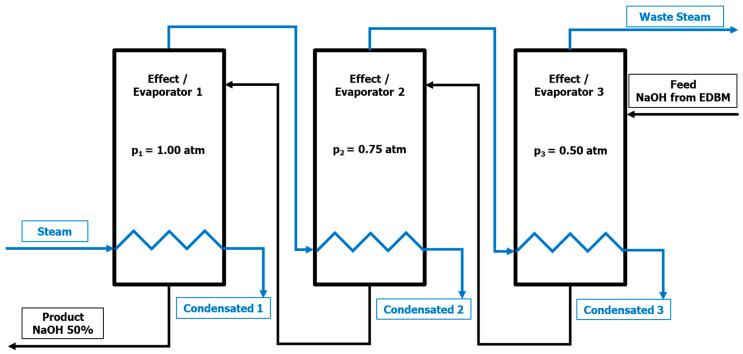
Flow diagram of the triple effect evaporation for the NaOH concentration.

**Figure 2 membranes-12-00885-f002:**
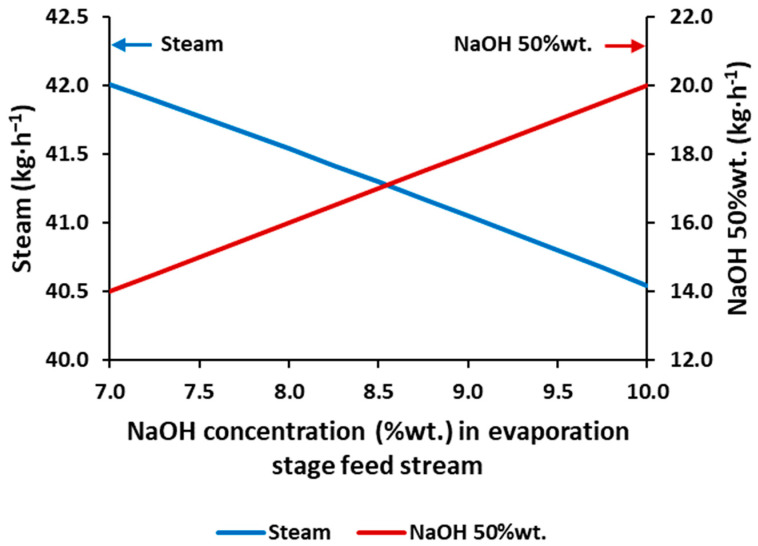
Industrial steam consumption and NaOH 50%wt. flowrate production versus the NaOH concentration in the feed stream of the simulated triple effect evaporation.

**Figure 3 membranes-12-00885-f003:**
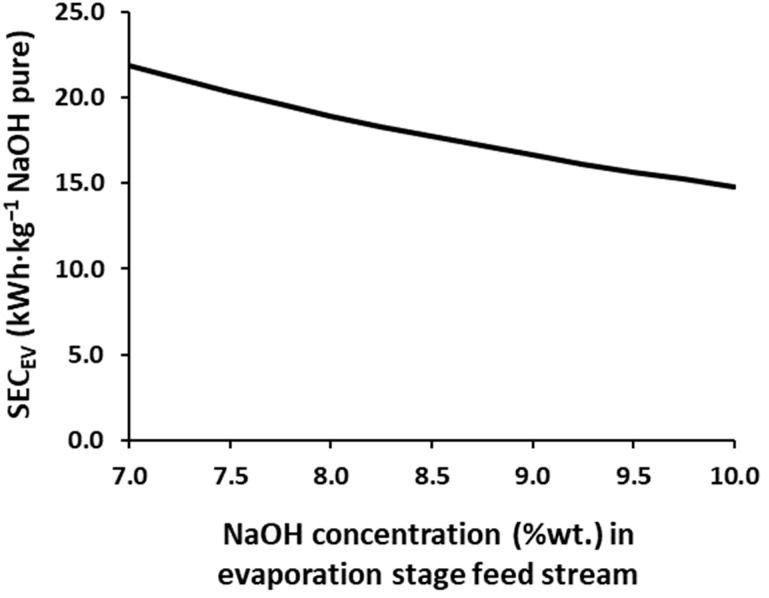
SEC_EV_ for different the NaOH concentrations in the feed stream of the simulated triple effect evaporation.

**Figure 4 membranes-12-00885-f004:**
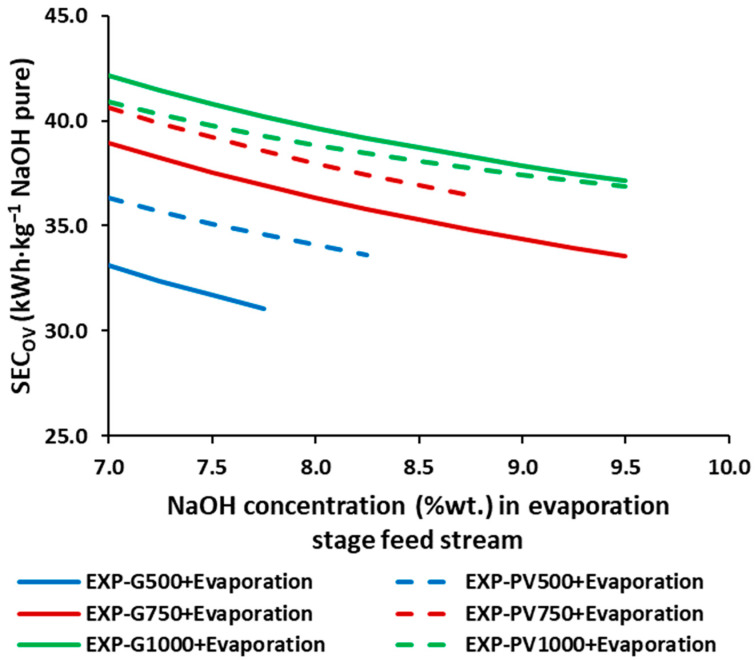
SEC_OV_ for the different NaOH concentrations in the feed stream of the simulated triple effect evaporation for a target final concentration of 50%wt. NaOH.

**Figure 5 membranes-12-00885-f005:**
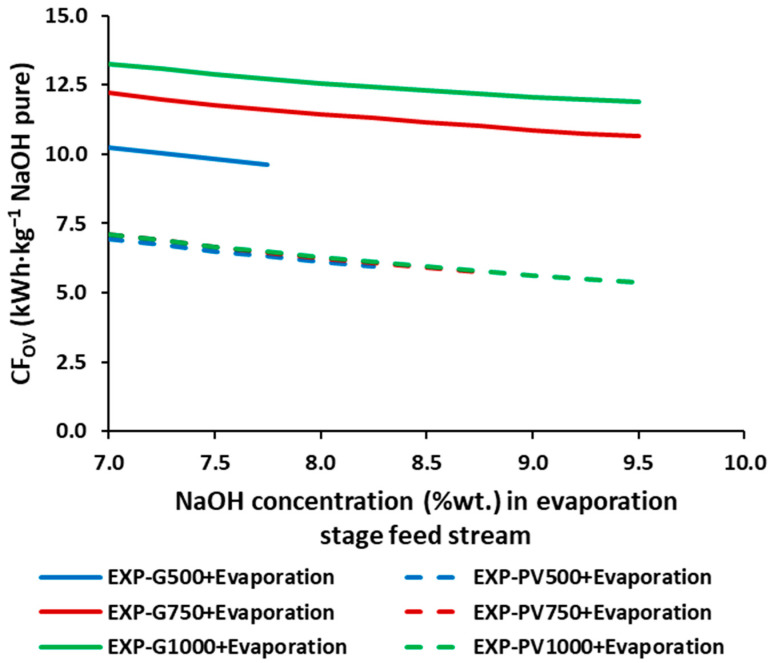
CF_OV_ for the different NaOH concentrations in the feed of the stream of the simulated triple effect evaporation.

**Table 1 membranes-12-00885-t001:** Summary of the maximum NaOH concentrations and the SEC_EDBM_. Adapted from Herrero-Gonzalez et al. [26].

Code	Energy Source	Current Density	NaOH		SEC_EDBM_
		A·m^−2^	mol·L^−1^	%wt.	kWh·kg^−1^ NaOH
Exp-G500	Grid Mix	500	2.95	7.81	11.6
Exp-G750	Grid Mix	750	3.63	9.63	16.5
Exp-G1000	Grid Mix	1000	3.63	9.63	23.4
Exp-PV500	Solar PV	500	3.10	8.21	15.5
Exp-PV750	Solar PV	750	3.34	8.84	20.2
Exp-PV1000	Solar PV	1000	3.65	9.66	22.6

**Table 2 membranes-12-00885-t002:** SEC_EDBM_ and the SEC_EV_ for different NaOH concentrations in the feed of the evaporation system for a target final concentration of 50%wt. NaOH.

NaOH in Feed (%wt.)	SEC_EDBM_ (kWh·kg^−1^ NaOH)						SEC_EV_ (kWh·kg^−1^ NaOH)
	Exp-G500	Exp-G750	Exp-G1000	Exp-PV500	Exp-PV750	Exp-PV1000	
7.00	11.2	17.1	20.2	14.4	18.7	19.0	21.9
7.25	11.3	17.1	20.4	14.6	18.8	19.2	21.1
7.50	11.4	17.2	20.5	14.8	18.9	19.5	20.3
7.75	11.4	17.3	20.6	15.0	19.0	19.7	19.6
8.00	-	17.4	20.7	15.1	19.0	19.9	18.9
8.25	-	17.5	20.9	15.3	19.1	20.1	18.3
8.50	-	17.6	21.0	-	19.2	20.3	17.7
8.75	-	17.6	21.1	-	19.3	20.5	17.2
9.00	-	17.7	21.2	-	-	20.8	16.6
9.25	-	17.8	21.4	-	-	21.0	16.1
9.50	-	17.9	21.5	-	-	21.2	15.7
9.75	-	-	-	-	-	-	15.2
10.00	-	-	-	-	-	-	14.8

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
