# Peer review of "Technical and Environmental Feasibilities of the Commercial Production of NaOH from Brine by Means of an Integrated EDBM and Evaporation Process"

_membranes, 2022, doi:10.3390/membranes12090885_

Round 1

Author Response

Authors appreciate the comments and suggestions by the reviewer.

Modifications regarding the structure and re-write of some sections has been carried out in order to improve the quality of the paper.

Here are the specific answers to the reviewer:

  1. As mentioned in the manuscript, both HCl and NaOH are produced simultaneously in the EDBM unit, however, both products are obtained at concentrations below the commercial. This fact presents the challenge of the integration of EDBM with other technologies in order to concentrate the products, each product its specific technology: i) distillation for HCl, and ii) evaporation for NaOH. Although previous works have reported the obtention of HCl at commercial concentrations by a combined process of EDBM and azeotropic distillation, there is a lack of study regarding the obtention of NaOH at commercial concentrations from SWRO brines. So, the main novelty of this study is the obtention of NaOH at commercial concentration from SWRO brines by means of a combined process of EDBM and a triple effect evaporation. Moreover, higher amounts of NaOH than HCl are consumed on desalination facilities, together with a higher market price. Thus, under authors consideration, it could be useful to evaluate both the technical and environmental feasibility of the combined process. The authors have highlighted the novelty of the work in the revised manuscript.
  2. Typos in Table 1 have been corrected.
  3. Authors have realized after the suggestions of the reviewers that the EDBM results were confusing. A brief description of the experimental methodology has been added to the methodology section, including how averaged current density when solar PV energy is applied is calculated.
  4. The composition of the condensate is 100% water. The description of the composition of the condensate has been included in the manuscript.
  5. As highlighted by the reviewer, the SECEDBM reported in Table 2 for the same product concentrations (e.g., 7%) are not equivalent for the different average current densities studied. Although higher current densities can lead in shorter operation times for the EDBM to reach a concentration value, this time is not shortened as much it is increased the energy input due to the fact of non-ideal phenomena such as concentration polarization, proton leakage or counter diffusion. Moreover, as higher is the concentration in the acid and base compartments, greater is the contribution of non-ideal phenomena in the performance of EDBM. As this work reports product concentrations higher than the commonly found in literature for EDBM, the influence of this non-ideal phenomena could be easily observed. Nevertheless, previous works of the authors demonstrated that higher current densities are required in order to obtain more concentrated products, even if higher SECEDBM are reported [1]. This higher currents densities are required in order to face the additional resistance that the non-ideal phenomena generate. EDBM technology is still limited by the product concentration obtained together with its high SEC, however author’s consider that the research efforts that are being carried out on the development of membranes could face this limitations.

References

  1. Herrero-Gonzalez, M.; Diaz-Guridi, P.; Dominguez-Ramos, A.; Irabien, A.; Ibañez, R. Highly concentrated HCl and NaOH from brines using electrodialysis with bipolar membranes. Sep. Purif. Technol. 2020, 242, 116785, doi:10.1016/j.seppur.2020.116785.

Reviewer 2 Report

Thanks, the paper is well written, however please check the following suggestions:

Re-write the abstract and conclusion; it is not well written 

Introduction part discuss some relevant literature 

Figure quality must be improved 

Thanks  

Author Response

Thank you very much for your nice comments.

Modifications regarding the structure and re-write of some sections has been carried out in order to improve the quality of the paper.

Here are the specific answers to the reviewer:

  • Authors have re-write abstract and conclusions.
  • Additional information has been added to the introduction.
  • Figures have been saved in a better-quality format and slightly modified in order to be improved and clearer to the reader.

Reviewer 3 Report

I recommend the publication of the manuscript Technical and environmental feasibility of commercial NaOH production from brines by means of an integrated EDBM and evaporation process with minor corrections, the authors have to eliminate some typos and inconsistencies:

Line 16: instead of "reduce"  it could be: "reducing";

I suggest to rephrase the whole paragraph between lines 59-63, to make it clearer.

Line 66: instead of "requires 0.2-0.5 g·m-3 of freshwater of HCl and 30-60 g·m-3 of freshwater of NaOH", I suggest: "requires concentrations of HCl of 0.2-0.5 g·m-3 of freshwater and of NaOH of 30-60 g·m-3 of freshwater"

Line 91: "variables" instead of "variable"

Line 95: to replace "standard commercial" with ” commercial standard concentration"

Figure 1: to correct the typo in "condesated"

Line 201: I suggest to replace "without the dependence on the product maximum concentration achieved" with " without dependence on the maximum concentration achieved for the product"

Author Response

Thank you very much for your comments.

Modifications regarding the structure and re-write of some sections has been carried out in order to improve the quality of the paper.

Typos and inconsistencies have been eliminated as indicated by the reviewer.

Round 2

Reviewer 1 Report

All my concerns have been addressed.